# Integrated Methodology for Potential Landslide Identification in Highly Vegetation-Covered Areas

Liangxuan Yan [1] , Quanbing Gong [1], Fei Wang [2], Lixia Chen [3], Deying Li [1] and Kunlong Yin [1],*

1   Faculty of Engineering, China University of Geosciences, Wuhan 430074, China
2   Zhejiang Academy of Geology, Hangzhou 310000, China
3   School of Geophysics and Geomatics, China University of Geosciences, Wuhan 430074, China
*   Correspondence: yinkl@cug.edu.cn

**Abstract:** It is normally difficult to identify the ground deformation of potential landslides in highly vegetation-covered areas in terms of field investigation or remote sensing interpretation. In order to explore a methodology to effectively identify potential landslides in highly vegetation-covered areas, this paper established an integrated identification method, including sliding prone area identification based on regional geological environment analysis, target area identification of potential landslides in terms of comprehensive remote sensing methods, and landslide recognition through engineering geological survey. The Miaoyuan catchment in Quzhou City, Zhejiang Province, southeastern China, was taken as an example to validate the identification methods. Particularly, the Shangfang landslide was successfully studied in terms of comprehensive methods, such as geophysical survey, drilling, mineral and chemical composition analysis, and microstructure scanning of the sliding zone. In order to assess the landslide risk, the potential runout of the Shangfang landslide was evaluated in a quantitative simulation. This paper suggests a methodology to identify potential landslides from a large area to a specific slope covered by dense vegetation.

**Keywords:** landslide identification; potential landslide; high vegetation coverage; southeast China





## 1. Introduction

Landslides are a common natural hazard that cause serious casualties and economic losses every year [1,2]. Effective prevention and reduction of landslide risk becomes a big challenge [3]. In order to reduce the landslide risk, there is a lot of research work on landslide identification, landslide susceptibility [4], landslide risk assessment, and landslide management [5,6]. Among these, the important work is accurate identification of landslides, especially in the highly vegetation-covered areas, because it is essential to identify landslides area and to deal with them efficiently before their failure.

Landslide identification is mainly conducted through field investigation, remote sensing techniques, and other methods. Engineering geological unit dominated the basic characteristics of landslides. A weak geological layer and slope structure provide an important and effective basis for landslide identification in landslide field surveys [7,8]. Remote sensing recognition has developed from optical remote sensing (ORS) to radar remote sensing. Previously, ORS relied on aerial photography. Since the Landsat satellite was launched into orbit in 1972, ORS has developed rapidly and is now mature [9,10]. At present, technologies such as big data and deep learning are explored and applied [11,12]. This has significantly enhanced the interpretation and continuously improved the automation of landslide identification. Synthetic aperture radar (SAR) is an all-weather active sensor. It uses microwaves that can penetrate clouds and fog for imaging. It can be widely used in areas that are difficult to image with ORS sensors. On the basis of SAR, the interferometric SAR (InSAR) was developed, which can perform high-precision observation of the ground surface [13,14]. Progressing from this, D-InSAR [15,16], PS-InSAR [17], SBAS-InSAR [18],

and other SAR observation technologies have been used widely in landslide investigation and monitoring [19–21]. InSAR has been verified in the identification of many landslides, such as the Baige landslide on the Jinsha River [22] and landslides in Pakistan [23].

However, in high vegetation coverage areas, visible light and microwaves cannot effectively penetrate the ground vegetation, and it is difficult to obtain actual surface morphology information. In addition, such areas often comprise high mountains and dense forests, which cause obvious difficulties in field investigation. Therefore, effectively identifying landslides in these areas is difficult. The current research on early identification of landslides is focused on the processing and interpretation of various images, paying less attention to geological environment analysis. As a result, there are many misjudgments and omissions in the interpretation of results. This phenomenon is especially significant in highly vegetation-covered areas. Therefore, effectively identifying landslides in these areas is difficult.

In this study, an integrated landslide identification method is innovatively proposed, which includes the classification of landslide prone areas based on geological environment background analysis, the identification of landslide target areas based on comprehensive RS, and the confirmation of landslide based on engineering geological survey. The workflow of an integrated methodology for potential landslide identification is suggested. Moreover, it further emphasizes the importance of engineering geological analysis in landslide identification. The Miaoyuan catchment in Quzhou City, Zhejiang Province, southeast China, was used to conduct landslide identification research in a high vegetation coverage area. This catchment has dense vegetation and abundant rainfall. It has been affected by landslides and suffered casualties in the past. Through our landslide identification methodology for highly vegetation-covered areas, potential landslides in this area were successfully identified, which is useful for further land planning. This method can improve the identification accuracy of potential landslides and be used in other high mountain areas with dense vegetation.

## 2. Materials and Methods

Different methods of landslide investigation and identification have their own characteristics and applicability, which make it difficult for a single method to achieve a useful result in the early identification of landslides in areas with high vegetation coverage [24]. Therefore, this study proposes an integrated set of potential landslide identification methods for high vegetation areas, which include prone sliding area identification based on regional geological environment analysis, target area identification of potential landslides in terms of comprehensive remote sensing methods, and landslide recognition through engineering geological survey (Figure 1). These methods comprise the analysis of geological conditions, the identification and evaluation of landslide pre-failure evidence, as well the simulation of landslide runout.

### 2.1. Landslide Prone Area Delineation

A landslide is defined as the failure of a rock mass on a slope, which involves the downward and forward movement of the damaged rock mass [25]. There are several conditions for landslides, including slope and weak interface. Generally, landslides are prone to and frequently occur in places with complex and rugged terrain. In addition, engineering construction projects will greatly change the topography. Many of these will trigger the occurrence of landslides. The weak interface can be the structural surface of a rock mass such as a bedding layer, joint surface, or fault. In areas with a complex geological stress history, faults, folds, and other structures are well developed, which weaken the strength of the rock mass and form dense joints. Due to the combination of joints and layers with the topography, dip slopes are extremely conducive to the occurrence of landslides. Water migration is another important triggering factor for landslides [26,27]. Moreover, water migration can be divided into multiple types such as extreme rainfall [28,29] and reservoir water storage [30].

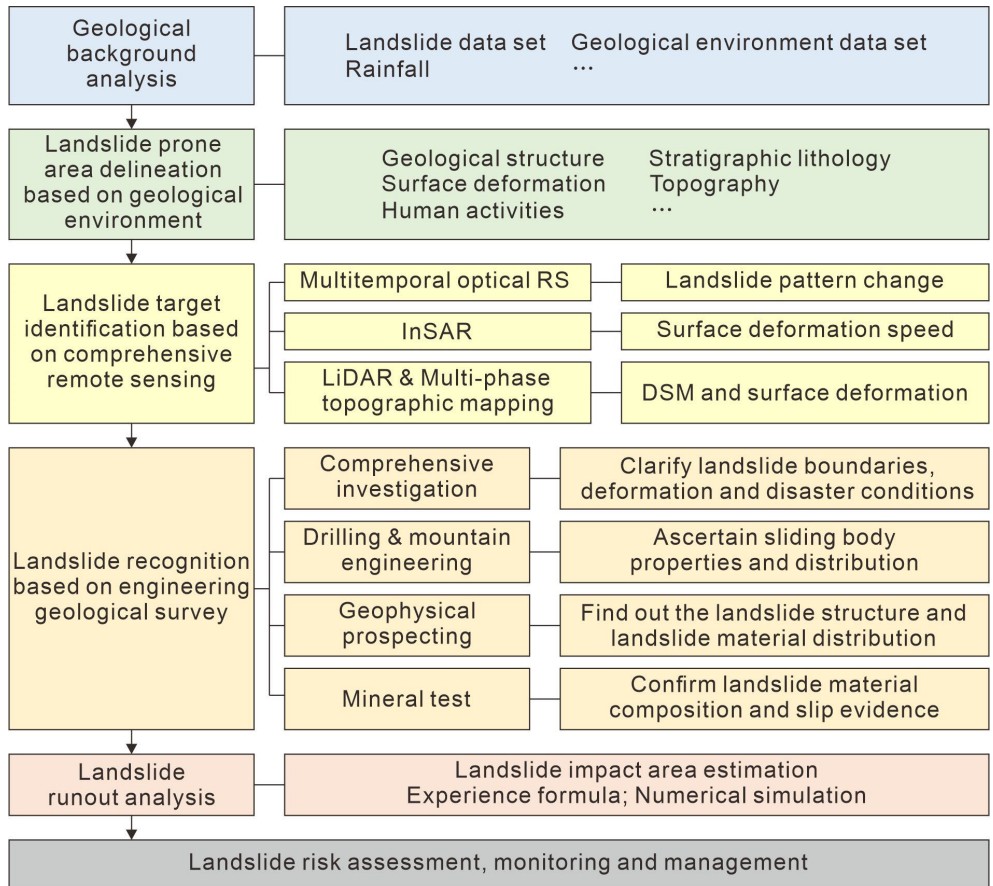

**Figure 1.** Landslide identification process in highly vegetation-covered area.

The above factors were analyzed to identify concealed landslides in the study area. The division criteria of the dominant potential landslide areas in the study area were proposed, as shown in Table 1.

**Table 1.** Landslide predomination area delineation condition.

| Elements | Characterization of Landslide Predomination Area |
|---|---|
| Topography | The intersection of different geographic units. Greatly fluctuating terrain. Slope with a large height difference. |
| Strata and Slope structure | Layered strata or massive stratigraphic units with developed structures. Dip slopes controlled by structural planes (layers, joints, faults). Formation with weak interlayers. |
| Geological structure | A history of many periods of tectonic stress, especially stress history from compression to extension. The junction of the core and the wing of the anticline. The contact zone of two structural or stratigraphic units. Intense fault and joint zone. |
| Deformation | Large-scale and multistage tectonic movement history. Slope deformation features, such as tensile cracks, scarp, gullies, tilled trees, etc. |
| Hydrogeology | Abundant rainfall or intensive rainfall. Rainfall convergence on slope surfaces. River erosion of the foot of slope. |
| Human activity | Road or building excavation at the toe of a slope. Reservoir construction, etc. |

### 2.2. Landslide Target Area Identification

Based on the analysis of the geological environment, comprehensive RS identification is conducted on the delineated prone areas to further analyze the possibility of potential landslides. The methods include ORS interpretation, InSAR, light detection and ranging (Li-DAR), multiphase topographic mapping, and digital image correlation (DIC). By combining these methods, the target area for potential landslides can be effectively identified.

Multitemporal ORS uses images of multiple time nodes and compares the changes in the time series plots to analyze the conditions, the deformation, and results of the landslides. Compared with the single-time-phase ORS, this process dynamically observes landslides. InSAR uses microwave radar to obtain ground information. Through differential interference processing, it obtains precise ground elevation information, terrain variation, and deformation velocity, and the accuracy can reach the millimeter level [31,32]. Because of its technical characteristics, InSAR has the advantage of surface displacement observation over large areas. LiDAR laser beams can penetrate vegetation and reach the surface [33]. Thus, vegetation information can be stripped, and an actual surface model can be obtained. It determines the trend and variation of the topography by comparing the results of multiphase topographic mapping. This provides a target area for the identification of potential landslides.

### 2.3. Landslide Recognition

After delineating the potential landslide prone areas and identifying the target areas, engineering geological surveys are conducted to identify the landslides. The engineering geological investigation of the landslide target areas includes a comprehensive survey, drilling, trenching, geophysical exploration, and microscopic mineral experiments [34,35]. Comprehensive field surveys are based on field investigation and interviews. The landslide information, such as boundary conditions and deformation history, is ascertained. In the field investigation, the macroscopic deformations, such as tensional cracks, scarp, and other stepped topographical features, are easily identified. Then, engineering geological mapping is completed to provide basic data for landslide recognition and evaluation. Engineering geological exploration, especially drilling, obtains the vertical structure of the formation, which is useful to identify the possible level of the sliding surface [36]. Geophysical exploration detects the spatial distribution of strata around the landslide and the ground water conditions [37,38]. It also further provides information about the composition of landslide materials. Microscopic mineral testing of soil along slide zone is designed to analyze the mineral composition and changes [39,40]. At the same time, microscopic scanning can help find the evidence of scratches or the directional arrangement of minerals along the sliding zone, which is formed due to the repeated shearing of landslide movement. It assists in proving the existence of landslides and the intensity of landslide movement. Microscopic pictures of the sliding zone soil structure are obtained by scanning electron microscope (SEM) [41]. During the repeated sliding, the clay minerals in the sliding zone may be observed in a directional arrangement, along with scratches, micropores, and microcracks.

### 2.4. Landslide Runout Prediction

Identification of landslides in high vegetation cover areas should include the assessment of their possible runout prediction. For this assessment, many scholars classify them as a category of landslide hazard assessment [42]. However, from the perspective of risk assessment, after the landslide is identified, a quick and simple analysis of the landslide impact area can identify landslides that have implications for further investigation and research.

Predicting landslide impact space in advance provides an important reference for landslide emergency management. An empirical estimation based on landslide gradient and volume is the most widely used [43]:

$$D = a \times (V \times \tan\theta)^b \tag{1}$$

where $D$ is runout, $\theta$ is the average slope, $V$ is the landslide volume, a and b are statistical parameters. When the detailed parameters of the landslide are available, the impact area of the landslide can also be estimated by numerical simulation.

## 3. Shangfang Landslide Identification

The Miaoyuan catchment was selected as the study area for landslide identification. It is located in the southwest of Zhejiang Province, China (Figure 2a). It belongs to the subtropical monsoon climate zone, with abundant rainfall and very high vegetation coverage (Figure 2c). The topography of the catchment is fragmented, and the slopes are steep. The Miaoyuan catchment is a set of layers made of pyroclastic rocks and intrusive magmatic rocks (Figure 2b). They are susceptible to weathering and erosion, which provide a source for the formation of landslides.

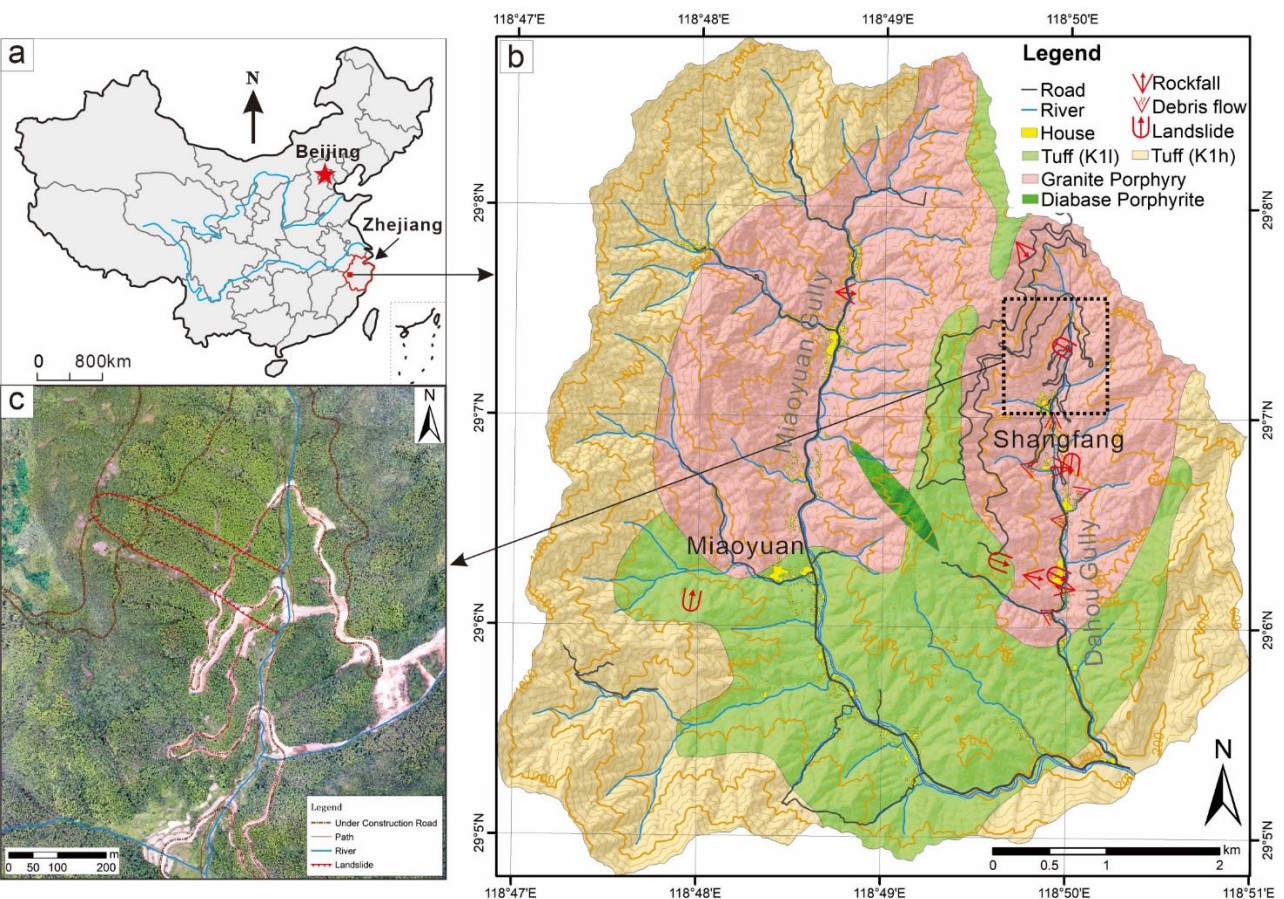

**Figure 2.** (**a**) The location of the Miaoyuan catchment. (**b**) The engineering geological map of the Miaoyuan catchment. (**c**) The digital orthophoto map (DOM) of a typical landslide (Shangfang landslide) in the Miaoyuan catchment.

### 3.1. Geological Environment Background

The study area is located in the Jiuhua Mountain area in the northern part of Quzhou City in the Jin-Qu Basin. It has undergone profound and prolonged tectonic evolution and formed complex and multiphased tectonic patterns (Figure 3a). The extrusion and

tension of historical stresses have greatly altered and changed the rock structure, resulting in a reduction in rock integrity and weakening rock strength. This has created favorable conditions for the development of landslides. Taking the Jiangshan–Shaoxing Fault (JSF) as the boundary, the study area is located in the northwestern part and belongs to the Yangtze basement. The study area was divided into five tectonic layers: the Jinning layer, Caledonian layer, Indosinian layer, Yanshanian layer, and Himalayan layer. From the old to the new, the area has experienced five stages of geological activities, including the Jinning ocean–continent subduction, the Caledonian land–continent collision collage stage, the Indosinian intracontinental orogenic stage, the Yanshanian ocean–continent subduction stage, and the Himalayan tectonic uplift stage. Figure 3b reflects the uplifting characteristics of Jiuhua Mountain during the latest tectonic movement stage. These tectonic activities have formed a complex and changeable terrain. The steep slope provides positive spatial conditions for the formation of landslides. The study area is the Jiuhua granite mass, which is the core part of the Jiuhua dome-shaped volcanic structure (Figure 3c). The area is directly controlled by the Changshan–Lizhu Fault (CLF). The Jiuhua pluton is medium–fine-grained porphyritic granite ($\gamma K_1$), which is in intrusive contact with the main part. The volcanic dome-shaped periphery is mainly composed of rhyolitic sintered tuff ($K_1 h$). The CLF is a regional fault that controls the tectonic pattern of the study area. It runs obliquely through the southern mountainous area of Jiuhua and strikes in the NEE (50~60°–230~240°), with a total length of about 250 km. The study area is located in the southern segment of the CLF, which consists of a series of parallel-trending brittle faults. In general, the study area comprises a set of brittle hard rocks of granite and tuff. Repeated compression and tension caused by tectonic movement have greatly reduced the strength of the rock mass. They form a rich source of material, which is conducive to the occurrence of landslides. Therefore, judging from the geological environment conditions, landslides in the Miaoyuan catchment are likely, especially in areas with steep slopes.

### 3.2. Shangfang Landslide RS Identification

Multitemporal ORS interpretation, InSAR, and multi-stage precise topographic mapping were used to identify potential landslides in the study area.

In this multitemporal ORS interpretation, four images from the 1960s, 1970s, 2002 and 2020 were selected for analysis. The image information and interpretation results are shown in Table 2 and Figure 4. According to the results of the multitemporal ORS interpretation, direct mapping of landslide susceptibility was carried out (Figure 4e). Combining the geological structure information and susceptibility map, the landslide target area was identified by engineering geological analogy. It included two key target areas and seven common target areas.

**Table 2.** Multitemporal ORS dataset information and interpretation results.

| Image Time | Data Source | Resolution Ratio | Image Type | Number of Landslides |
|---|---|---|---|---|
| 1960s | KeyHole | 2.23 m | Panchromatic | 88 |
| 1970s | KeyHole | 2.23 m | Panchromatic | 58 |
| 2002 | KeyHole | 0.56 m | Panchromatic | 76 |
| 12 November 2020 | WorldView-2 | 0.50 m | True Color | 9 |

This study selected 36 Sentinel-1 SAR images from 4 May 2020 to 28 June 2021. SBAS-InSAR was used to analyze the surface deformation of the Miaoyuan catchment (Figure 5a). According to the interpretation results, the data points were concentrated in the bottom of two valleys. On the contrary, the data points on the hillsides were scarce because of dense vegetation, due to which there may be errors. In order to better perceive the deformation of in the catchment, we projected the deformation data onto the slope units. Thus, the deformation of the slope unit was obtained (Figure 5b). According to the results of InSAR interpretation, the slope on the west side of the catchment and the middle ridge had the largest deformation velocity, exceeding 90 mm/a. This trend was more obvious from the

deformation velocity of the slope unit. Accordingly, the northwest part of the catchment was divided into landslide target areas (Figure 5b). The central landslide target area has had landslides on 15 August 2002 (slope unit A) and 4 June 2020 (slope unit B).

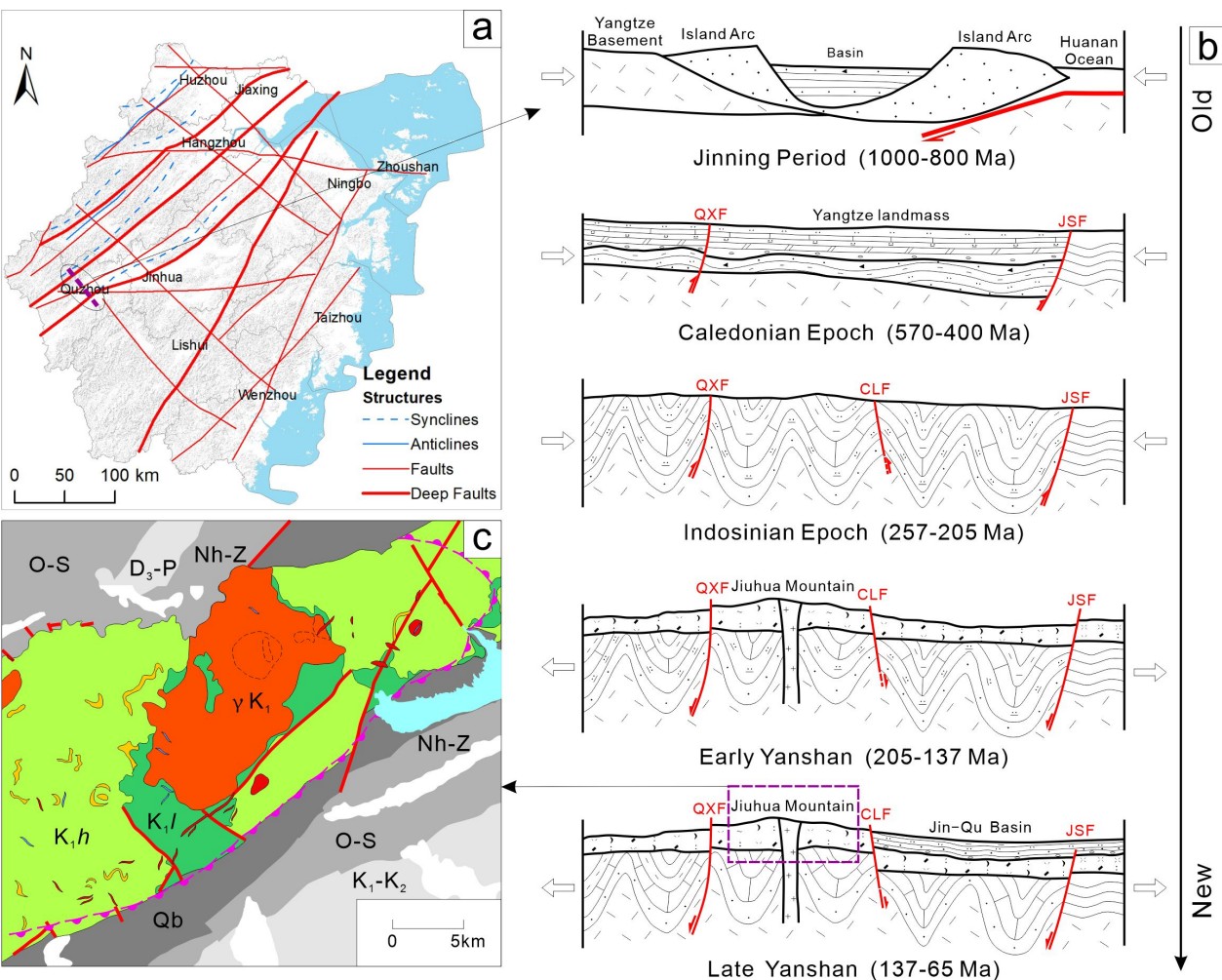

**Figure 3.** Diagram of regional geological structural evolution. (**a**) Structural outline map of Zhejiang Province. (**b**) Geological structural evolution of Jin-Qu Basin. (**c**) Geological and structural map of Jiuhua Mountain (QXF: Qiuchuan–Xiaoshan Fault).

The comparison of elevation from topographic maps is useful for the identification of slope deformation. In this study, the topographic map of 2012 at the scale of 1:10,000 and the topographic map of 2017 at the scale of 1:2000 were compared, using these two topographic maps to compare the elevation changes around the landslide (Figure 6). On the topographic difference map, the northern part mainly showed subsidence, and the central and southern parts were mainly uplifted. Combined with the landslide identified by RS, this showed a slumping sliding process of subsidence at the upper slope and uplift at the foot slope. It provided strong evidence for landslide identification in a high vegetation coverage area. If there are more episodes of terrain data, this change process will be more detailed.

### 3.3. Shangfang Landslide Structure and Mechanism

Detailed surveys were carried out in the target area identified by comprehensive RS. Taking the Shangfang landslide as an example, based on field investigation and geophysical exploration, drilling and other engineering geological surveys were carried out. The landslide was also confirmed by microstructure analysis.

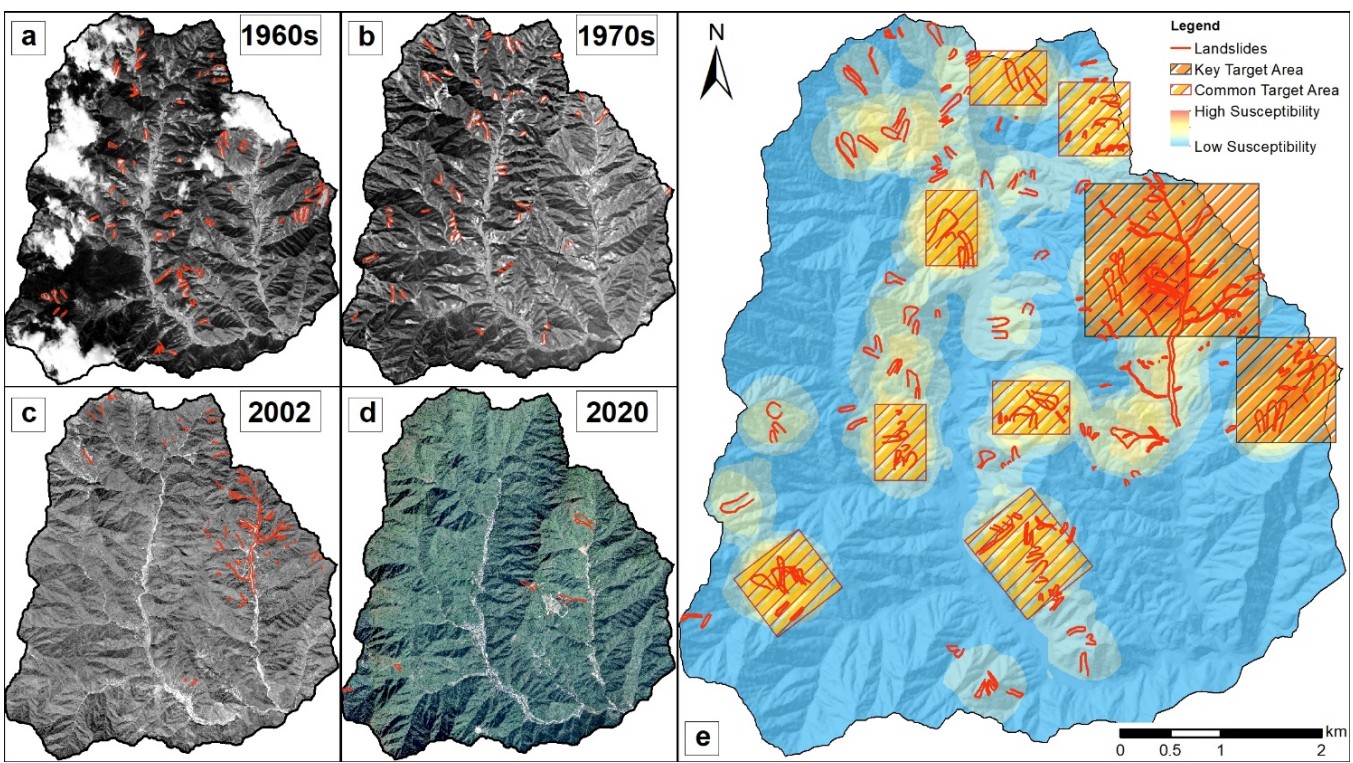

**Figure 4.** Interpretation results of multitemporal ORS: (**a**) 1960s, (**b**) 1970s, (**c**) 2002, and (**d**) 2020. (**e**) Landslide susceptibility direct mapping by multitemporal ORS interpretation.

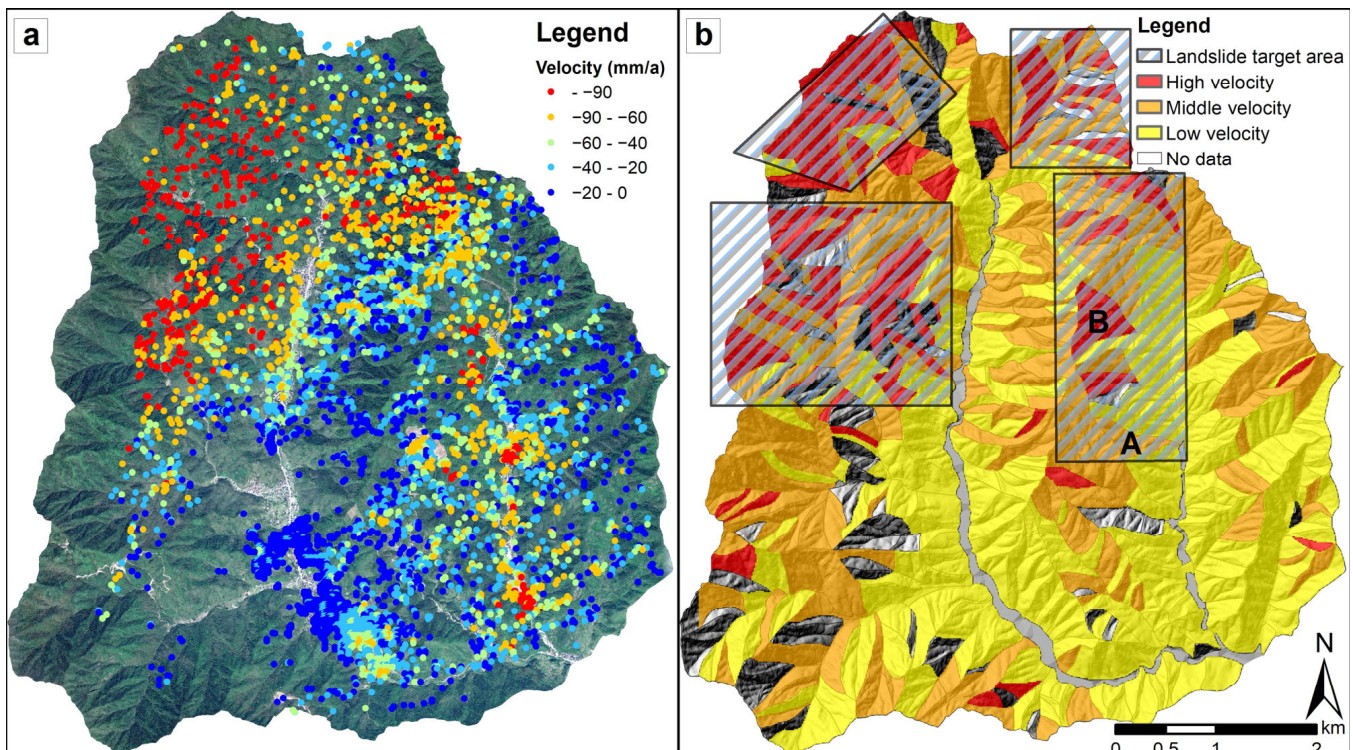

**Figure 5.** Deformation map of the Miaoyuan catchment. (**a**) Deformation map interpreted by InSAR, and displacement information is missing on the slope, (**b**) Deformation map of slope unit.

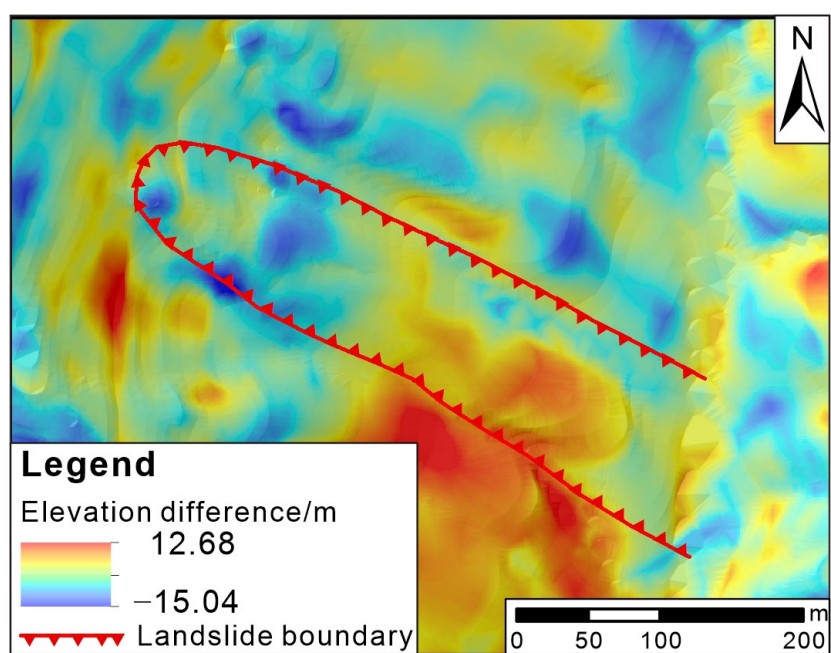

**Figure 6.** The elevation differences between the topographic mappings in November 2012 and March 2017.

The Shangfang landslide is located upstream of the Dahouyuan gully (Figure 2). The landslide is a soil landslide. The sliding direction is southeastern 115°. The elevation of the main scarp is 780 m, and the foot slope is 510–540 m. The landslide is 450 m long and 100 m wide, with an area of about 45,000 m$^2$. From the sections (Figure 7) of the multi-electrode resistivity method (MER) and shallow seismic reflection method (SSR), the average thickness of the landslide was estimated to be about 8 m and the volume about 330,000 m$^3$. The sliding surface of the landslide is steep on the top and gentle on the foot, with an average gradient of 28° (Figure 8). The lithology of the bedrock of the landslide is lightly–moderately weathered granite ($\gamma K_1$). Controlled by regional faults, NEE–SWW- and NNE–SSW-trending joints and fissures developed in the sliding bed. The right side of the landslide is bounded by a gully. The bedrock of the landslide was exposed on the bottom of the gully, but the left boundary is covered by weathered soil. There are pine trees on the upper part of the landslide and bamboo in the middle and lower parts, and the coverage rate is more than 90% (Figure 7). There is a historic temple on the top of the mountain, and the local tourism bureau planned a new road along the hillside. The newly built highway just cut the foot of the landslide and passed three times on the landslide body, which may greatly reduce the stability of the landslide. Meanwhile, there is also an electric power tower that belongs to the 10 kV Qingtai 8270 Power Line built on the landslide body.

According to the field investigation, the landslide is controlled by the discontinuities and faults in the rock mass. The sliding body is composed of brown silt and segmented granite grains. The content of segmented grain is more than 40%, and the size is 20–50 cm (Figure 9). The soil of the sliding zone is brown silty clay mixed with debris. The soil is loose and saturated (Figure 9b). There are parallel, arc-like arrangements of minerals due to the long-term shearing process on the sliding zone (Figure 9c). When the slip soil is uncovered, parallel scratches along the slip direction are seen (Figure 9d,e). This reflects the multiple displacements and shears that occurred during the sliding process of the landslide. Undisturbed soil was sampled from the slip soil (Figure 9c) for the microstructure test. Microstructure observation was performed using an FEI Quanta 200 SEM produced by the FEI Company (Figure 10h).

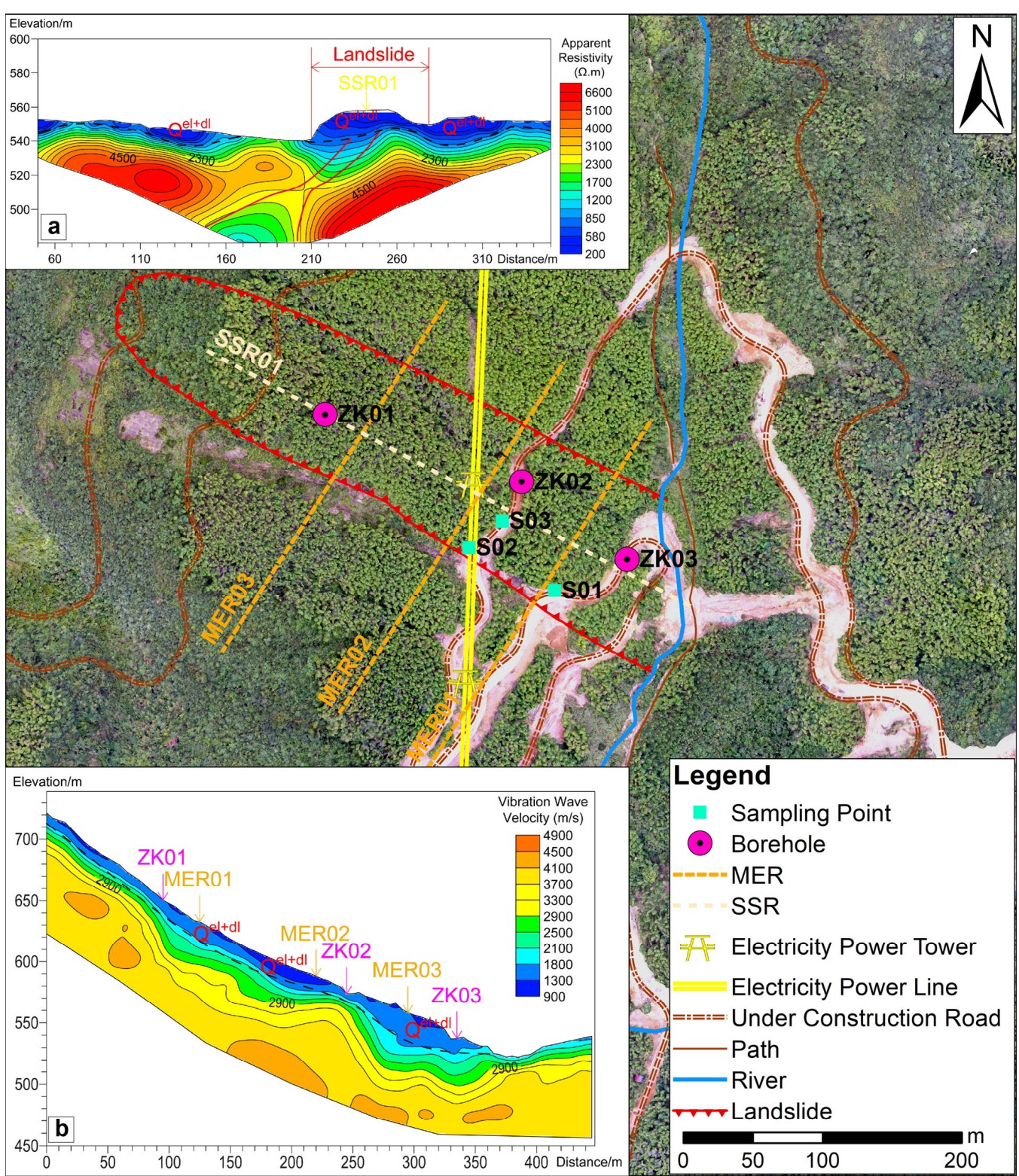

**Figure 7.** Shangfang landslide investigation plan. (**a**) SSR01 section. (**b**) MER01 section.

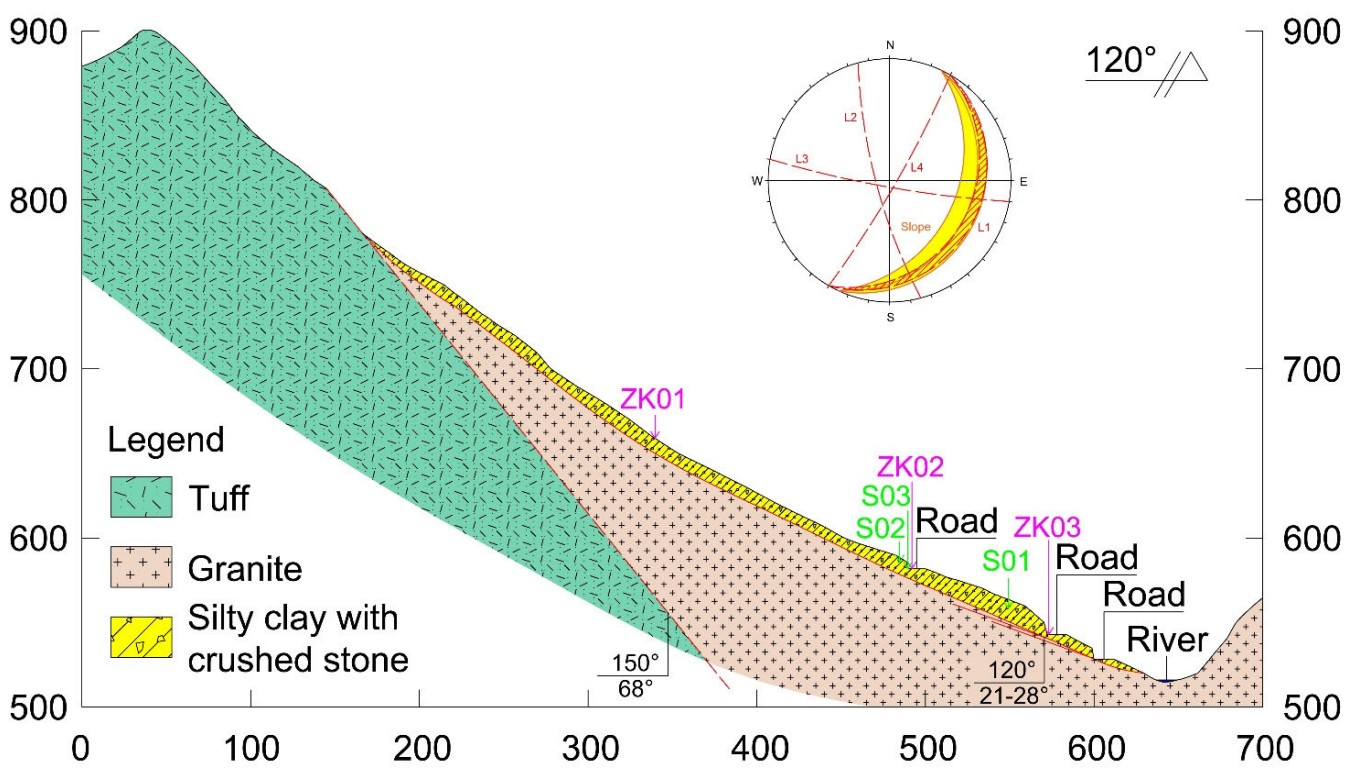

**Figure 8.** Shangfang landslide engineering geological profile.

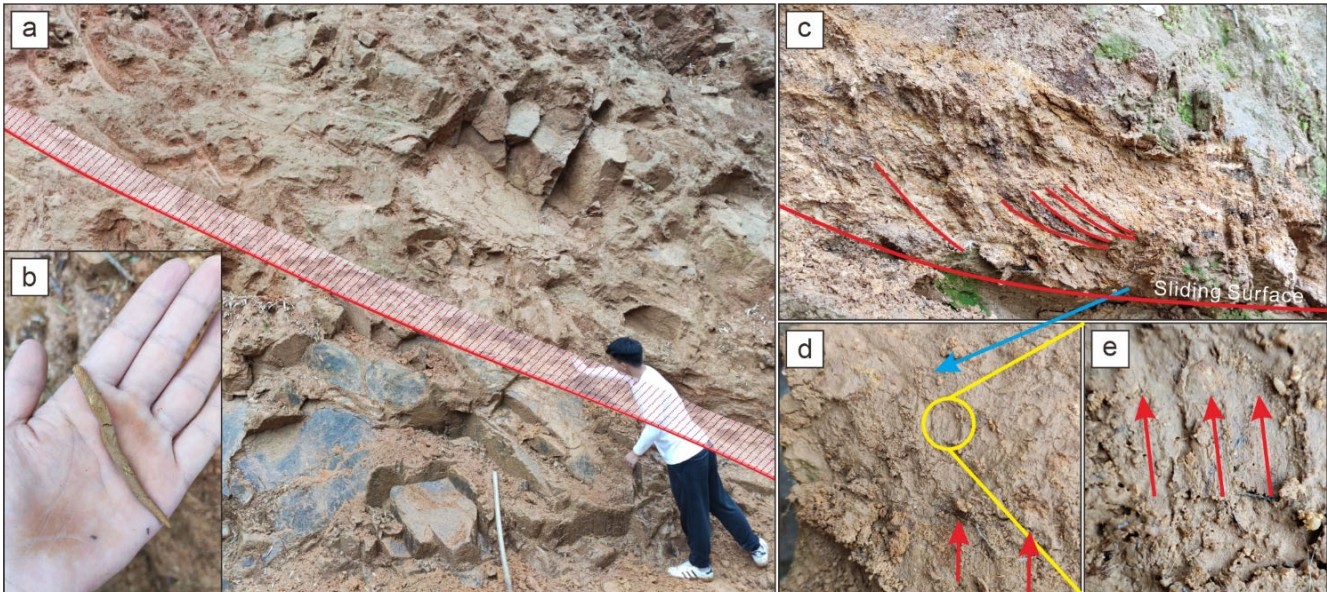

**Figure 9.** Slip soil photos at S1. (**a**) Sliding belt space distribution. (**b**) Slip soil rubbing test. (**c**) Arc-shaped arrangement of soil particles and a sliding surface. (**d**) Scratches on the sliding surface. (**e**) Enlarged view of scratches.

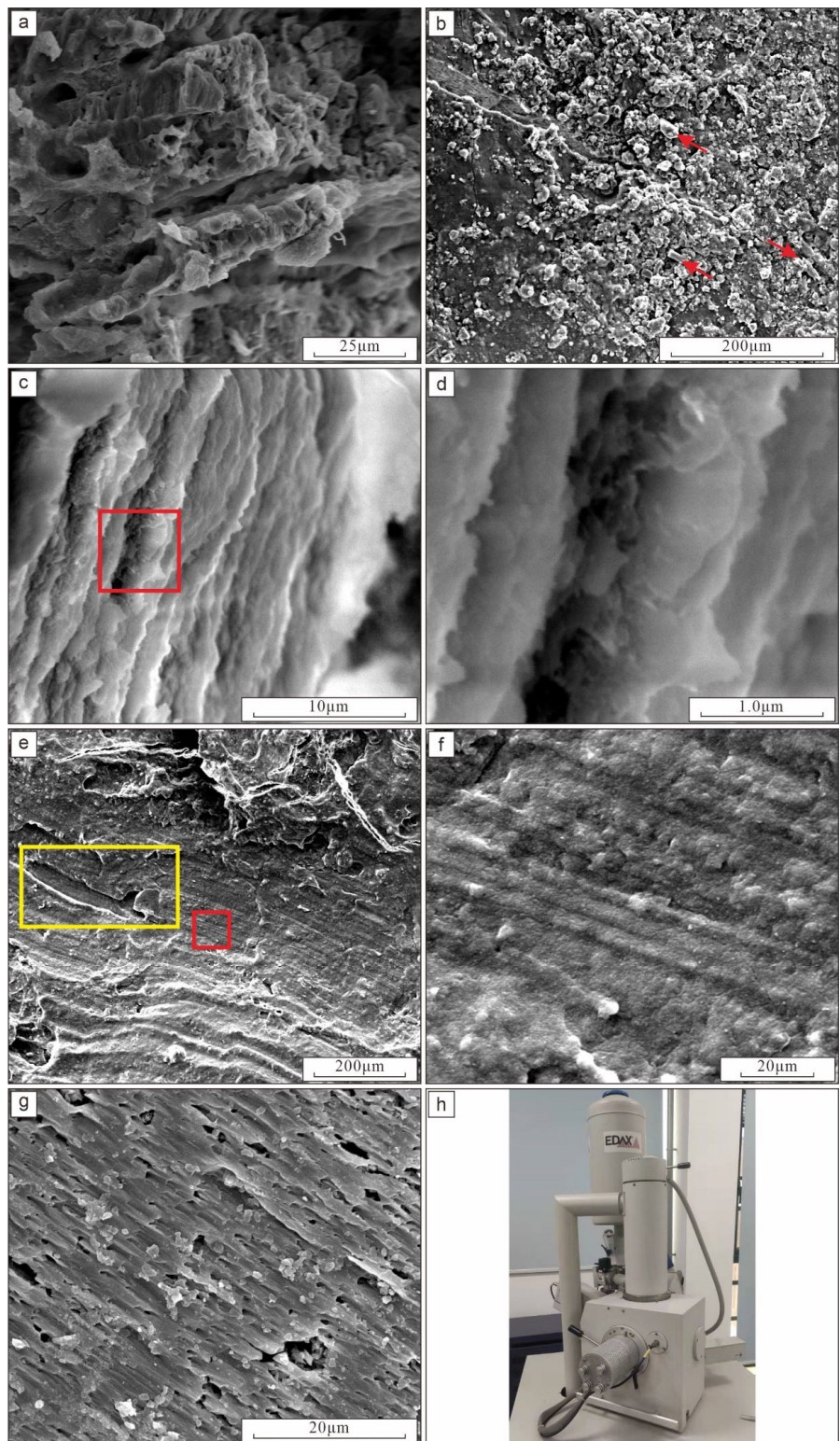

**Figure 10.** Microstructure of Shangfang landslide slip soil and the SEM. (**a**) Directional arrangement of flaky minerals (1200×). (**b**) Directional arrangement of mineral particles such as quartz (250×). (**c**) Layered arrangement of clay minerals (5000×). (**d**) Magnified image of layered arrangement of clay minerals (20,000×). (**e**) Arc scratches (yellow box) and straight scratches (red box) (150×). (**f**) Magnified image of straight scratches (1200×). (**g**) Leaching structure of granite (2500×). (**h**) FEI Quanta 200 SEM.

SEM observed parallel arrangement in the same direction of the clay minerals in the sliding zone and scratches, micropores, and microcracks (Figure 10). Because of compression and friction, the mineral particles are aligned and elongated, reducing the friction coefficient between particles, which can reduce the frictional strength of the sliding zone. During the process of landslide movement, the soil particles were sheared and gradually broken and decomposed into fine soil particles. Fine soil particles formed strips or sheets along the shear direction (Figure 10a). On the other hand, because of the effect of shear friction, as the shear displacement increased, the mineral particles were oriented along the shear direction (Figure 10b). For sheet-like clay minerals under repeated shearing and friction, a layered and stacked book-sheet-like structure was formed. Figure 10c shows the layered clay mineral structure observed from the direction perpendicular to the sliding surface, and Figure 10d is an enlargement of the red square in Figure 10c. The scratches are grooves formed by extruding and scraping soil particles or mineral particles on the sliding zone, sometimes with steps. They are generally in straight and arc shapes. Straight scratches reflect a relatively fast friction movement, which resulted from the fast sliding stage of the landslide. At this time, the slip soil is subjected to great shear stress, which pushes the mineral particles to move rapidly. The scratches are straight and deep, as shown in the red square in Figure 10e, which is enlarged in Figure 10f. Different scratch directions represent different shear stages and multiple sliding processes. The causes of arc-shaped scratches are complex. According to the phenomenon observed under the SEM, the movement of the sliding zone was blocked by large mineral crystals or soil particles, which resulted in the shifting of the movement path. The velocity of the landslide was slow; the local movement path was bent and the mineral particles rotated. The arc-shaped scratches were formed, with some shallow scraping grooves in the foot and deep ones in the head, as shown in the yellow square in Figure 10e. It is assumed to be caused by slow friction, indicating that the landslide was creeping. Micropores and microcracks are common in the sliding surfaces. Micropores result from the leaching of the sliding zone and the dissolution of minerals. These pores increase the porosity of the sliding zone. Once the slope is damaged because of shear, it stimulates a higher pore water pressure, which is an important formation mechanism of landslides. As shown in Figure 10g, this is the leaching structure of the granite sliding zone. The easily weathered mica and potassium feldspar form a secondary pore because of the rapid weathering rate and the leaching of groundwater. After rainfall infiltration, the pores are saturated with water, and the landslide tends to slide. The excess pore water pressure increases and the effective stress decreases, which increases the possibility of the landslide. These pores have an obvious orientation. Their direction is consistent with the direction of the landslide seepage field.

Water flows along rock fissures and fractured zones in weathered granite, with the decomposition of feldspar. The hydrolysis and ion exchange reactions of silicate minerals are important factors in the origin of clay minerals. The hydrolysis of primary silicate minerals is mainly caused by $H^+$. A highly concentrated source of $H^+$ is provided by the ionization of water and the presence of acid. The hydrolysis of minerals is mainly due to the $Na^+(K^+)$–$H^+$ exchange, which is hydration (Figure 11). Feldspar hydrolyzes under the action of dissolved filtrate. Dissolution pits develop along the feldspar cleavage plane, bicrystal junction plane, and dislocation and other structurally weak planes. This is a crystal surface alteration phenomenon caused by kaolinization in the early weathering stage of feldspar. The solution filtrate continuously washes the feldspar, and various shapes of dissolution pits, such as prismatic and honeycomb-shaped pits, penetrate deep beneath the surface of the feldspar crystal to the inside. The connection and merging of these dissolution pits forms grooves (Figure 10g). The structure of feldspar is destroyed, and the mineral is transformed into kaolin. The book-like or lamination-like kaolinite polycrystals and pompon-like tubular halloysite aggregates are formed (Figure 10c).

**Figure 11.** The process of feldspar hydrolysis [44].

Three samples were analyzed for mineral component changing with Innov XRD-Terra (Table 3): the sliding zone sample (SF 01-1), the bed rock sample (SF 01-2), and the sliding body sample (SF 02). The test results are shown in Table 4 and Figure 12. For the Shangfang landslide, the content of albite decreased significantly, and the kaolinite content increased significantly from the bed rock sample (SF 01-2) to the sliding belt sample (SF 01-1). This reflected the massive decomposition of albite during the weathering evolution from bed rock to slip zone. Some minerals were removed during groundwater seepage by leaching, resulting in the increase of the content of quartz. Because of the increase in the kaolinite content, the shear strength of the sliding surface deceased, especially the internal friction angle. This was also consistent with the results of the landslide soil mechanical parameters test. For the sliding body sample (SF 02), the composition was complex because the evolution of the sliding body was affected by multiple stages of collapse and accumulation from the upper slope. As inferred from the quartz content, the dissolution and filtration of the sliding body were more intense. A large portion of the clay minerals was removed by groundwater, so their content was less. This indicated that the weathering of the sliding body was more intense. Combined with the geotechnical test results, because of the increase in the quartz content, the internal friction angle of the sliding body was higher than that of the slip belt soil.

**Table 3.** Introduction to Innov XRD-Terra.

| Equipment | Innov XRD-Terra | |
|---|---|---|
| Voltage | 30 kV | |
| Current | 300 μA | |
| Target | Cu | |
| Step width | 0.02° | |
| Vibration frequency | 6000 Hz | |
| Angle range | 5–55° (2θ) | |

**Table 4.** The content of each mineral in the samples.

| Sample No. | Position | Phases Contained in the Sample (%) | | Crystallinity (%) |
|---|---|---|---|---|
| SF 01-1 | Sliding belt | Quartz | 21.9 | 78.8 |
| | | Albite | 44.5 | |
| | | Kaolinite | 18.8 | |
| | | Clinochlore | 18.8 | |
| SF 01-2 | Sliding bed | Albite | 70.8 | 78.9 |
| | | Quartz | 8.6 | |
| | | Illite | 11.9 | |
| | | Kaolinite | 8.6 | |
| SF 02 | Sliding body | Lizardite | 11.0 | 88.4 |
| | | Augite | 12.8 | |
| | | Quartz | 37.6 | |
| | | Kaolinite | 14.4 | |
| | | Chlorite | 24.3 | |

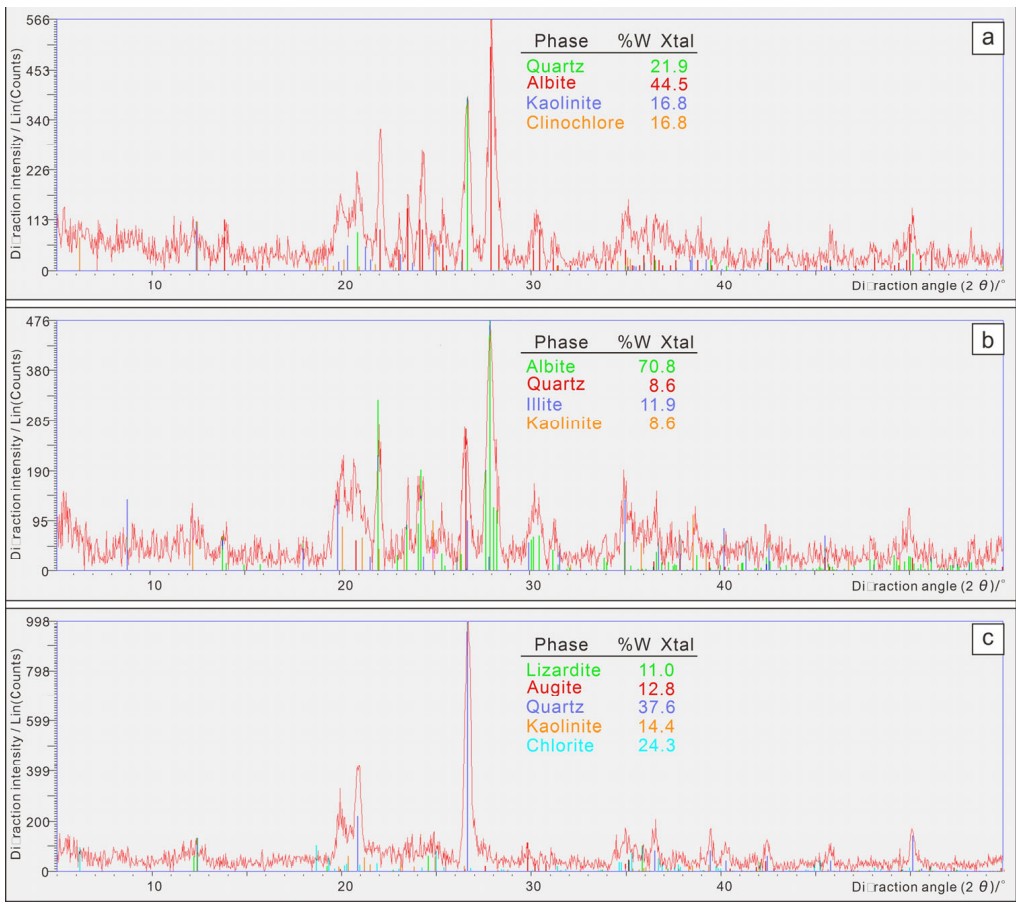

**Figure 12.** Characteristic peak D value and fitting spectrum of each phase. (**a**) XRD pattern of sample SF 01-1, (**b**) XRD pattern of sample SF 01-2, (**c**) XRD pattern of sample SF 02.

In the study area, through the analysis of the geological environment, we determined that the Miaoyuan catchment is a landslide prone area with rugged terrain and a complex geological structure. Multitemporal ORS interpretation, InSAR, and multiphase topographic mapping all explained that landslides were likely and frequent in the top part of Dahouyuan gully. In the catchment, nine target areas were delineated by multitemporal ORS, and four target areas were delineated by InSAR. Two of them highly overlapped, so there were eleven landslide target areas in total. After the engineering geological survey, 15 landslides were finally confirmed as shown in Figure 2b. Taking the Shangfang landslide as an example, through a series of engineering geological surveys and laboratory test analysis of the target area, the landslide was finally confirmed. The landslide was caused by the rapid uplift of the Jiuhua Mountain area since the Yanshan period. The deep ravine created favorable terrain conditions. Because of the influence of tectonic stress, the surface granite porphyry was cut, and its weathering process was accelerated. The group of joints approximately parallel to the slope formed the continuous bedrock interface. Multiple sets of scratches observed at the site indicated that creep may occur many times during the landslide displacement. The Shangfang landslide was also characterized by topographic features. The front of the landslide was convex and squeezed the river (Figure 7). On 15 August 2002, the right side of the Shangfang landslide slipped triggered by heavy rainfall.

## 4. Shangfang Landslide Impact Area Identification

The Shangfang landslide was recently discovered by our research project in this catchment. The landslide body is uninhabited. There was no historical record of sliding deformation. However, according to the RS and field investigation, the landslide has

occurred with several periods of movement. In the foot part and the middle of the landslide, a tourist road construction cut the sliding body at different elevations and formed new free slopes, which affected the stability of the landslide. Rainfall is the main triggering factor. Cut slopes have local failures during intensive rainfall periods. The tendency of the landslide is unstable. There are potential risks to people and vehicles on the road after construction is finished. If the landslide slips, the risk is great.

According to the landslide early warning studies issued by the administrative department, the highest level of warning threshold value for rainfall in this area is 180 mm of accumulated effective rainfall in five days. The mechanical parameters of landslide stability analysis were determined by engineering geological survey and test, as shown in Table 5. At the highest warning level, the failure probability of the landslide exceeds 50%. Landslide runout analysis was simulated. The DAN3D software platform was used to simulate the movement process of the landslide, as shown in Figure 13. According to the simulation results, the landslide will interrupt the tourist road and damage the power tower (011 s in Figure 13). Moreover, it will block the Dahouyuan gully and affect the local tourism industry. This study has proposed our suggestion to local administration agencies for further landslide risk management.

**Table 5.** Shangfang landslide stability calculation parameters.

| $\gamma$ (kN/m³) | | c (kPa) | | $\varphi$ (°) | |
|---|---|---|---|---|---|
| | Mean | Standard Deviation | Mean | Standard Deviation | |
| Natural state | 19.0 | 18.6 | 2.5 | 30.7 | 2.9 |
| Saturation state | 20.5 | 16.5 | 2.2 | 27.4 | 2.7 |

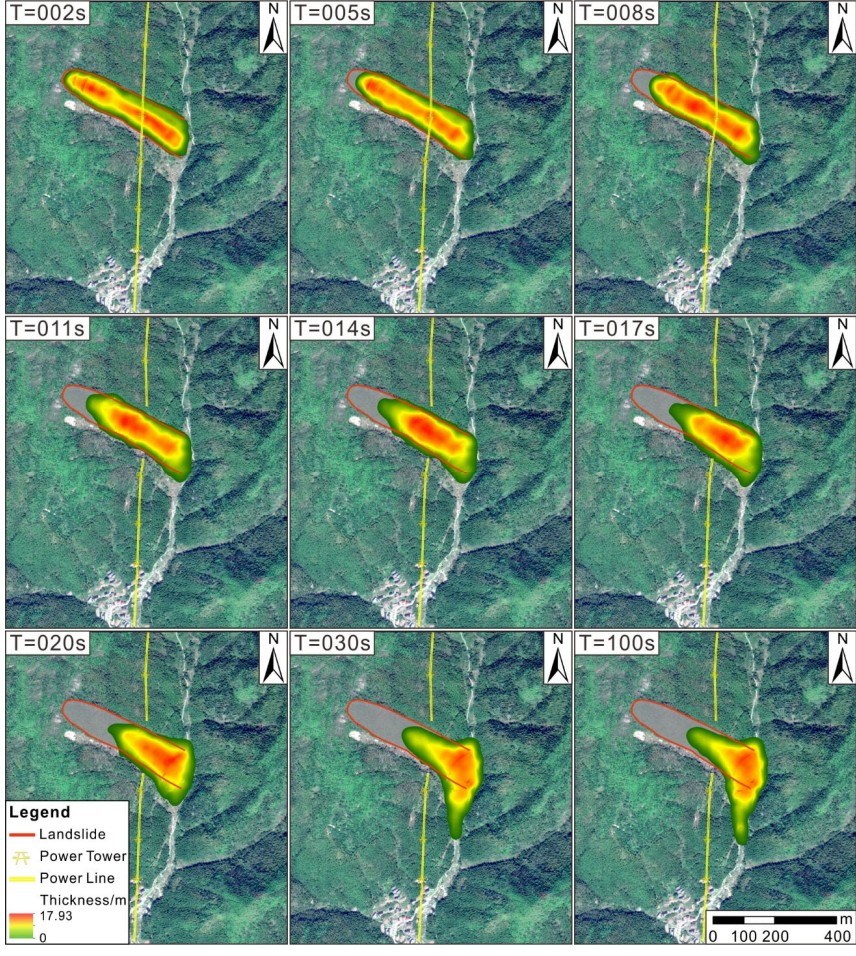

**Figure 13.** Shangfang landslide movement process.

## 5. Discussion

In highly vegetation-covered areas, ORS interpretation can only detect landslides that have slipped or have significant deformation. Potential landslides and slightly deformed landslides are difficult to identify because of the vegetation cover. Therefore, there are often many omissions in the landslide interpretation results obtained directly from ORS. Similarly, most of the current SAR satellites represented by Sentinel-1 are equipped with C-band synthetic aperture radar, which has difficulty penetrating vegetation. As a result, they tend to be out of coherence in areas of high vegetation cover or fail to interpret the data (Figure 5a). For multiphase topographic surveying and mapping, because of the errors of surveying and mapping themselves in mountainous area, the topographical elevation difference may be enlarged; it will cause big deviations and omissions. The results cannot be used directly. Therefore, comprehensive judgment is useful by analyzing regional engineering geological conditions combined with the identification of target areas of landslide by RS. A single method of RS recognition can only focus on the partial deformation characteristics of potential landslides. Therefore, some landslides may not be identified, or non-landslide areas with similar characteristics may be identified as landslides. The comprehensive RS identification guided by the landslide target area avoids the misjudgment and missed judgment caused by a single method. The potential landslides were identified from geological environment conditions, RS features, and landslide deformation evidence. For example, the landslide prone areas can be found from the analysis of landform and tectonic evolution history. The landslide target area is identified by RS methods such as ORS and InSAR to integrate the change of landslide patterns and terrains. The evidence chain of landslide is further improved through engineering geological survey and sliding belt soil test. Therefore, this method can comprehensively use these methods to identify landslides, and the potential landslides can be finally confirmed accurately.

It is also important to evaluate the runout of the identified landslides for the purpose of risk assessment, which is linked to the elements at risk. This is the entire framework from identification of landslide to risk assessment. Objectively, there are a large number of landslides in nature. It is impossible to investigate and evaluate all landslides with limited human resources. Therefore, the assessment of landslide runout distance serves to identify the relevant landslides that are dangerous to the elements at risk. Thus, it can improve the work efficiency of landslide identification.

## 6. Conclusions

This study proposed an integrated methodology of landslide identification for highly vegetation-covered areas. The methodology included the landslide prone area delineation based on the geological environment analysis, the landslide target area identification based on comprehensive RS interpretation, and landslide recognition based on an engineering geological survey. Taking the Miaoyuan catchment in the Quzhou City, southwest of Zhejiang Province as an example, the potential landslides represented by the Shangfang landslide were identified. For this landslide, a geological background analysis, comprehensive RS interpretation, engineering geological survey, and mineral composition and microstructure analysis were carried out. These formed a flowchart of the landslide identification from a region to a specific slope. Meanwhile, runout analysis of the landslide was conducted to assess the impact range of potential landslides toward the elements at risk, in order to improve the work efficiency of landslide identification. The integrated methodology can effectively improve the recognition of potential landslides and be used in other areas with dense vegetation, such as the southeast and southwest of China and the Three Gorges Reservoir area.

**Author Contributions:** Conceptualization, L.C.; Methodology, L.Y. and K.Y.; Investigation, L.Y., Q.G., F.W. and K.Y.; Resources, L.C.; Data curation, D.L.; Writing—original draft, L.Y.; Writing—review & editing, D.L. and K.Y.; Visualization, Q.G.; Supervision, K.Y.; Project administration, F.W. All authors have read and agreed to the published version of the manuscript.

**Funding:** This research was funded by Kecheng Branch of Quzhou Natural Resources and Planning Bureau grant number ZZCG2021058.

**Conflicts of Interest:** The authors declare no conflict of interest.

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
