# Peer review of "Integrated Methodology for Potential Landslide Identification in Highly Vegetation-Covered Areas"

_remotesensing, doi:10.3390/rs15061518_

Round 1

Reviewer 1 Report

This paper established a comprehensive identification method, including prone sliding area identification based on the regional geological environment analysis, target areas identification of potential landslides in terms of comprehensive remote sensing methods and landslide recognition. The work is highly innovative and of practical application. The theory is well described and the experiments are adequate, but the following problems remain.

(1)The meaning of a and V in equation (1) is not explained.

(2)Some of the figures are not clear enough, e.g. Figures 11 and 12, and the meaning of the horizontal and vertical coordinates and units are not given in Figure 12.

(3)The analysis in the Discussion section is not sufficient and should clarify the advantages of your proposed method over other methods and explain the corresponding advantages through what operations, and finally analyse the limitations of the proposed method.

(4) In the Conclusion section you need to include the potential application areas and prospects of your method.

Author Response

Dear reviewer:

We would like to thank you for your kind constructive comments concerning our manuscript previously entitled “Integrated methodology for potential landslide identification in highly vegetation-covered areas”. Those comments are valuable and helpful to improve this article. We have revised the manuscript in accordance, and uploaded the manuscript with the changes highlighted in red. The point-by-point responses are as follows:

Point 1: The meaning of a and V in equation (1) is not explained.

Response 1: We have supplemented the explanations of ,  and .  is the landslide volume,  and  are statistical parameters. (lines 157-158)

Point 2: Some of the figures are not clear enough, e.g. Figures 11 and 12, and the meaning of the horizontal and vertical coordinates and units are not given in Figure 12.

Response 2: We modified Figure 11 (line 348) and Figure 12 (line 371), supplemented the horizontal and vertical coordinates, and provided a high-definition version.

Point 3: The analysis in the Discussion section is not sufficient and should clarify the advantages of your proposed method over other methods and explain the corresponding advantages through what operations, and finally analyse the limitations of the proposed method.

Response 3: We further analyzed and expounded the technical advantages of our method. It reduces the false recognition rate and missed recognition rate of landslide, and determines the potential landslide according to the geological environment conditions, remote sensing characteristics and landslide deformation evidence. Moreover, landslide identification from the risk perspective can improve the identification efficiency. The details are as follows:

A single method of RS recognition can only focus on the partial deformation characteris-tics of potential landslides. Therefore, some landslides cannot be identified or non-landslide areas with similar characteristics will be identified as landslides. The com-prehensive RS identification guided by the landslide target area avoids the misjudgment and missed judgment caused by a single method. The potential landslides were identified form geological environment conditions, RS features and landslide deformation evidence. For example, the landslide prone areas can be found from the analysis of landform and tectonic evolution history. Landslide target area is identified by RS methods such as ORS and InSAR to integrate the change of landslide pattern and terrain. The evidence chain of landslide is further improved through engineering geological survey and sliding belt soil test. Therefore, this method can comprehensively use these methods to identify landslides. The potential landslides are finally confirmed accurately. (lines 429-440)

Objectively, there are a large number of landslides in nature. It is impossible to investigate and evaluate all landslides with limiting human resources. Therefore, the assessment of landslide runout distance is the identification of landslide for the elements at risk. It can identify the landslides that are dangerous to the elements at risk. Thus, it can improve the work efficiency of landslide identification. (lines 443-448)

Point 4: In the Conclusion section you need to include the potential application areas and prospects of your method.

Response 4: We added the potential application areas and prospects of the method. We believe that the integrated methodology can effectively improve the recognition of potential landslides and be used in other areas with dense vegetation, such as the southeast and southwest of China and the Three Gorges Reservoir area. (lines 461-463)

Reviewer 2 Report

Comprehensive methodology for potential landslide  identification in highly vegetation-covered areas

This MS has the good potential to publish in the Remote Sens. after minor revision. It focuses on a Comprehensive methodology for potential landslide identification in highly vegetation-covered areas. Conventionally used techniques are adapted. The authors perform a very good job and this is a very good contribution to the geoscience community.

Some Suggestions

Title

It is suggested to remove the word Comprehensive methodology from the title and replace it with an appropriate word like integrated methodology

Introduction

The section needs minor improvement. However, the sentences are not correctly interconnected for an easy flow. The introduction part does not give a clear explanation of the innovation and significance of the paper. Please modify it and write comprehensively.

*       What are your key results?

*       What do you recommend as a further study for others in the same study area?

Line 25-29 it is suggested to revise the writing comprehensively.

Line 112 2.3 Landslide recognition

This section needs to provide proper references

3.1 Geological environment background

How geology is helpful to identify Landslides?

Author Response

Dear reviewer:

We would like to thank you for your kind constructive comments concerning our manuscript previously entitled “Integrated methodology for potential landslide identification in highly vegetation-covered areas”. Those comments are valuable and helpful to improve this article. We have revised the manuscript in accordance, and uploaded the manuscript with the changes highlighted in red. The point-by-point responses are as follows:

Point 1: Title

It is suggested to remove the word Comprehensive methodology from the title and replace it with an appropriate word like integrated methodology.

Response 1: We have replaced the comprehensive methodology with integrated methodology, include the title and body (lines 2, 12, 59, 63, 77, 450, 461).

Point 2: Introduction

The section needs minor improvement. However, the sentences are not correctly interconnected for an easy flow. The introduction part does not give a clear explanation of the innovation and significance of the paper. Please modify it and write comprehensively.

What are your key results?

What do you recommend as a further study for others in the same study area?

Line 25-29 it is suggested to revise the writing comprehensively.

Response 2: We have completely revised the introduction section to improve the language and logic, especially the original 25-29 lines. (lines 25-72)

The innovation of this article is: The innovative combination of geological environment background analysis, comprehensive remote sensing and engineering geological survey forms the methodology and workflow of integrated methodology for potential landslide identification in high-vegetation areas.

The key results are: An integrated landslide identification method was innovatively proposed, which includes the classification of landslide prone areas based on geological environment background analysis, the identification of landslide target areas based on comprehensive RS and the confirmation of landslide based on engineering geological survey. (lines 59-62)

We recommend it is need to further emphasize the importance of engineering geological analysis in landslide identification. (lines 63-65)

Point 3: Line 112 2.3 Landslide recognition.

This section needs to provide proper references

Response 3: We have supplemented references on landslide comprehensive survey [34,35], drilling [36], geophysical exploration [37,38], mineral composition analysis [39,40] and microstructure testing [41]. (lines 125-146)

  1. Palenzuela, J.A.; Jiménez-Perálvarez, J.D.; El Hamdouni, R.; Alameda-Hernández, P.; Chacón, J.; Irigaray, C. Integration of LiDAR Data for the Assessment of Activity in Diachronic Landslides: A Case Study in the Betic Cordillera (Spain). Landslides 2016, 13, 629–642, doi:1007/s10346-015-0598-x
  2. Chelli, A.; Mandrone, G.; Truffelli, G. Field Investigations and Monitoring as Tools for Modelling the Rossena Castle Landslide (Northern Appennines, Italy). Landslides 2006, 3, 252–259, doi:1007/s10346-006-0046-z.
  3. Borrelli, L.; Antronico, L.; Gullà, G.; Sorriso-Valvo, G.M. Geology, Geomorphology and Dynamics of the 15 February 2010 Maierato Landslide (Calabria, Italy). Geomorphology 2014, 208, 50–73, doi:1016/j.geomorph.2013.11.015.
  4. Li, D.; Yan, L.; Wu, L.; Yin, K.; Leo, C. The Hejiapingzi Landslide in Weining County, Guizhou Province, Southwest China: A Recent Slow-Moving Landslide Triggered by Reservoir Drawdown. Landslides 2019, 16, 1353–1365, doi:1007/s10346-019-01189-5.
  5. Bichler, A.; Bobrowsky, P.; Best, M.; Douma, M.; Hunter, J.; Calvert, T.; Burns, R. Three-Dimensional Mapping of a Landslide Using a Multi-Geophysical Approach: The Quesnel Forks Landslide. Landslides 2004, 1, 29–40, doi:1007/s10346-003-0008-7.
  6. Perrone, A. Lessons Learned by 10 Years of Geophysical Measurements with Civil Protection in Basilicata (Italy) Landslide Areas. Landslides 2021, 18, 1499–1508, doi:1007/s10346-020-01584-3.
  7. Regmi, A.D.; Yoshida, K.; Dhital, M.R.; Devkota, K. Effect of Rock Weathering, Clay Mineralogy, and Geological Structures in the Formation of Large Landslide, a Case Study from Dumre Besei Landslide, Lesser Himalaya Nepal. Landslides 2013, 10, 1–13, doi:1007/s10346-011-0311-7.
  8. Wang, F.; Zhang, S.; Li, R.; Zhou, R.; Auer, A.; Ohira, H.; Dai, Z.; Inui, T. Hydrated Halloysite: The Pesky Stuff Responsible for a Cascade of Landslides Triggered by the 2018 Iburi Earthquake, Japan. Landslides 2021, 18, 2869–2880, doi:1007/s10346-021-01656-y.
  9. Chen, J.; Dai, F.; Xu, L.; Chen, S.; Wang, P.; Long, W.; Shen, N. Properties and Microstructure of a Natural Slip Zone in Loose Deposits of Red Beds, Southwestern China. Engineering Geology 2014, 183, 53–64, doi:1016/j.enggeo.2014.10.004.

Point 4: 3.1 Geological environment background

How geology is helpful to identify Landslides?

Response 4: The complex tectonic evolution and stress stage formed rugged terrain and destroyed the structure of rock mass. They provide topographic and material conditions for the occurrence of landslides. We analyzed this in combination and determined the landslide prone area in the study area. (lines 186-188, 199-201)
